# Influence of the Fluoroscopy Setting towards the Patient When Identifying the MPFL Insertion Point

**DOI:** 10.3390/diagnostics12061427

**Published:** 2022-06-09

**Authors:** Alexander Korthaus, Tobias Dust, Markus Berninger, Jannik Frings, Matthias Krause, Karl-Heinz Frosch, Grégoire Thürig

**Affiliations:** 1Department of Trauma and Orthopaedic Surgery, University Medical Center Hamburg-Eppendorf, 20246 Hamburg, Germany; a.korthaus@uke.de (A.K.); t.dust@uke.de (T.D.); m.berninger@uke.de (M.B.); j.frings@uke.de (J.F.); m.krause@uke.de (M.K.); k.frosch@uke.de (K.-H.F.); 2Department of Trauma Surgery, Orthopaedics and Sports Traumatology, BG Hospital, 20246 Hamburg, Germany

**Keywords:** MPFL, fluoroscopic MPFL identification, MPFL attachment, MPFL reconstruction

## Abstract

(1) The malposition of the femoral tunnel in medial patellofemoral ligament (MPFL) reconstruction can lead to length changes in the MPFL graft, and an increase in medial peak pressure in the patellofemoral joint. It is the cause of 36% of all MPFL revisions. According to Schöttle et al., the creation of the drill canal should be performed in a strictly lateral radiograph. In this study, it was hypothesized that positioning the image receptor to the knee during intraoperative fluoroscopy would lead to a relevant mispositioning of the femoral tunnel, despite an always adjusted true-lateral view. (2) A total of 10 distal femurs were created from 10 knee CT scans using a 3D printer. First, true-lateral fluoroscopies were taken from lateral to medial at a 25 cm (LM25) distance from the image receptor, then from medial to lateral at a 5 cm (ML5) distance. Using the method from Schöttle, the femoral origin of the MPFL was determined when the femur was positioned distally, proximally, superiorly, and inferiorly to the image receptor. (3) The comparison of the selected MPFL insertion points according to Schöttle et al. revealed that the initial determination of the point in the ML5 view resulted in a distal and posterior shift of the point by 5.3 mm ± 1.2 mm when the point was checked in the LM25 view. In the opposite case, when the MPFL insertion was initially determined in the LM25 view and then redetermined in the ML5 view, there was a shift of 4.8 mm ± 2.2 mm anteriorly and proximally. The further positioning of the femur (distal, proximal, superior, and inferior) showed no relevant influence. (4) For fluoroscopic identification of the femoral MPFL, according to Schöttle et al., attention should be paid to the position of the fluoroscopy in addition to a true-lateral view.

## 1. Introduction

The medial patellofemoral ligament (MPFL) is the most essential passive protection against lateral patellar dislocation within the first degrees of knee joint flexion [1]. Therefore, MPFL reconstruction in patients with patellar instability is the established basic procedure, which can be complemented by other procedures [2,3]. Most studies report good to excellent clinical outcomes and low redislocation rates [4]. However, other articles report remarkable complication rates (up to 26% ± 11) [5]. Biomechanical studies have shown that the native MPFL is nearly isometric at 0 to 110 knee joint flexion [6,7]. Length changes of the reconstructed MPFL are most sensitive to malpositioning of the femoral tunnel, especially in the proximal–distal axis [6,7,8].

Intraoperative fluoroscopy is the gold standard for guiding anatomic and isometric femoral tunnel placement [9,10,11,12,13,14,15]. An exact true-lateral alignment of the femoral condyles during fluoroscopy is mandatory [16,17].

Femoral tunnel misplacement is seen in 31% to 67% of cases undergoing MPFL revision surgery [10,18,19]. A recent systematic review reported femoral tunnel misplacement as the leading reason in 38.2% of revision cases [20]. This result indicates that placement remains challenging.

Misplacement of the femoral tunnel can lead to non-isometric and non-physiologic patella guidance during knee flexion, thus creating non-physiological contact pressure between the patella and trochlea [6,8]. 

Previous work identifying the MPFL by fluoroscopy has always positioned the operated knee in the middle of the C-arm, which is not always consistent with practice. The fluoroscopy of three-dimensional objects always shows a size distortion due to object-to-image-receptor distance (OID). The parts of the object that are farther away from the image receptor are radiographically represented with more significant size distortion than the parts of the object that are closer to the image receptor [21]. Even if the object is in close contact with the image receptor, some parts are farther away than others. The OID plays an essential role in minimizing the size distortion of the radiographic image [21]. It has not yet been investigated whether misplacement of the identified MPFL point occurs when the position is not centered to the image receptor despite a true-lateral view.

This study investigated the sensitivity of the femoral tunnel position under lateral fluoroscopy as a function of the position of the image receptor to the knee. It was hypothesized that positioning the image receptor to the knee would lead to a relevant mispositioning of the femoral tunnel in identifying the MPFL, despite an always adjusted true-lateral view.

## 2. Materials and Methods

This study used ten 3D printed polylactic acid (PLA) femurs from anonymized patients (mean age 43 years old; range 17–69). CT scans of these patients were saved as a complete DICOM series and processed using Materialise’s Interactive Medical Image Control System and 3-Matic (Mimics Innovation Suite v24; 3-Matic Medical v16; Materialize, Leuven, Belgium). Post-processing was performed with global surface treatment. For the slicing process, Cura (Ultimaker Cura v4.11; Ultimaker, Utrecht, The Netherlands) was utilized with a layer height of 0.1 mm and a gyroid infill structure to ensure a high level of detail and steady X-ray images. Then, 3D Printing was performed using an Ultimaker S5 Dual Head Fused Deposition Modeling (FDM) printer with polyvinyl alcohol (PVA) as a support material. None of the CT scans showed evidence of dysplasia according to the Dejour classification [22]. 

Each femur was fixed on a fixture. First, true-lateral images from lateral to medial were performed at a distance of 25 cm from the image receptor (LM25 view). Then, true-lateral images from medial to lateral were performed at a distance of 5 cm from the image receptor (ML5 view). A fluoroscopy (Ziehm Imaging GmbH, FPD 8 × 8 inch) was used (Figure 1 and Figure 2). Identification of the MPFL point was determined by the senior author (GT), an experienced orthopedic knee fellow.

The true-lateral view was defined as an image in which both the posterior medial and lateral femoral condyle (control for rotation) and the distal medial and lateral femoral condyle (control for adduction/abduction) were precisely aligned [16]. Native femoral insertion of the MPFL was assessed using a standardized 10-mm radiographic plate and MPFL identification template, according to the radiographic landmarks described by Schöttle et al. [23], which represent the anatomic placement of the femoral tunnel for MPFL reconstruction (Figure 3). 

The radiographic landmarks for femoral tunnel placement were also applied when the femur was positioned distal, proximal, superior, and inferior to the detector plate (Figure 4) without changing the position of the radiographic plate. 

Only those images were evaluated, which allowed a true-lateral image and an application of the MPFL identification template in the respective outermost position. Likewise, the MPFL was identified in a true-lateral position in the ML5 view and the LM25 view, respectively, and controlled in the true-lateral position in the LM25 view and the ML5 view, respectively (Figure 5). 

According to a biomechanical study [6,8], the critical distance of a tunnel was defined as having its center at least 5 mm from the center of the identified native MPFL, as mentioned above. Length measurements were made using a digital image archiving and communication system (Horos Version 3.3.6). The Horos system measurement tool was calibrated to the 10-mm plate to ensure the accuracy of the data and compensate for any magnification error.

Data are shown as mean ± SD. Differences were calculated using the Wilcoxon signed-rank test (2-related sample), and a *p* value of less than 0.05 was considered significant. All analyses were performed with the IBM SPSS Statistics version 26 program (IBM Corp., Armonk, NY, USA).

## 3. Results

The images in the outermost position (distal, proximal, anterior, and posterior) showed a significant change of 0.9 mm to 1.9 mm in their identified center of the MPFL, in both the LM25 and ML5 images (Table 1).

When the femoral MPFL insertion point identified in the ML5 view was verified in the LM25 view, a critical distance of 5.3 mm ± 1.2 mm resulted. The MPFL identified in the ML5 was primarily posterior, and distal to the Schöttle point in the LM25 view (Figure 6).

A near-critical distance of 4.8 mm ± 2.2 mm resulted when the femoral MPFL insertion point was identified in the LM25 view and verified in the ML5 view. The MPFL identified in the LM25 view was primarily anterior, and proximal to the Schöttle point in the ML5 view (Figure 7). 

## 4. Discussion

The main finding of this study was that the positioning of the image receptor to the operated knee (ipsilateral or contralateral) shows a relevant difference in identifying the femoral MPFL insertion point in the true-lateral view. The identified femoral MPFL insertion point in the ML5 view showed a difference of approximately 5 mm from the femoral MPFL insertion point identified in the control LM25 view. The same result was obtained with the femoral insertion point identified in the LM25 view.

In the last decade, MPFL reconstruction in patients with recurrent episodes has gained much importance and attention, and various surgical techniques have already been published [24,25,26,27,28,29,30,31,32,33]. Moreover, every study emphasizes the importance of correctly identifying the femoral footprint. A preferred type is the one proposed by Schöttle et al. [23], which requires a true-lateral view [16]. Clinical studies on MPFL reconstruction have shown “good” to “excellent” results to date [4]. However, femoral misplacement of the MPFL tunnel can lead to failure of MPFL reconstruction. In revision surgery, femoral MPFL tunnel malpositioning is the most common complication, at 38.2% [20].

Several authors have already shown that true-lateral views that are not well aligned show slight differences in the localization of the MPFL point already at 2.5°, and significant differences at 5° [16,17,34]. Howells et al. point out that a true lateral view with different rotations causes an error in anterior–posterior alignment [17]. Balcarek and Walde showed that a significant error could also occur with misaligned abduction and adduction [16]. They used six cadaveric femurs for this purpose, and determined the femoral MPFL point according to Schöttle et al. [23]. Ziegler et al. confirmed similar results, using dissected cadaver knees with markings on each anatomical structure [34]. These studies considered neither the eccentric position of the knee joint, nor the effect of OID, nor the image receptor’s position. This study analyzed the relevant fluoroscopic settings for identifying the MPFL insertion point in daily practice. The eccentric positioning of the knee for identifying the femoral MPFL footprint had no clinical impact. However, the positioning of the image receptor, and thus the influence of OID, showed a relevant effect on the localization of the femoral MPFL footprint.

The proper positioning of the femoral MPFL tunnel and the tolerance of its misplacement is still discussed in contradictory ways in the literature [9,10,18,35,36]. However, various biomechanical works agree that more than 5 mm misplacement leads to an increase in medial peak pressure in the patellofemoral joint [6,8,37]. These studies also show that non-isometric tunnel placement leads to an increase in graft stress during knee joint flexion. 

Rather than relying on percutaneous radiographic techniques to achieve the most anatomic MPFL reconstruction possible, an anatomic approach is suggested by some, in order to take into account individual anatomy [34,38,39,40]. Thus, accurate performance of an MPFL reconstruction requires an incision large enough to identify the essential anatomic landmark, the adductor tuberosity, and the medial epicondyle. This method proves more difficult in corpulent patients and could involve a large incision. Therefore, the authors recommend pre-surgical 3D CT reconstruction of the bone surface in these cases [38,41]. For current daily practice, however, the Schöttle technique [23] with adjustment [16,17] prevails in most cases. Various studies have shown that a smaller scatter is achieved than in anatomical identification [11,12,13].

The fluoroscopy of three-dimensional objects always shows a size distortion due to OID. The closer the object to the image receptor, the smaller the size distortion [21]. The position of the fluoroscopy to the patient must be changed in clinical practice to reduce the OID from 25 cm to 5 cm. An OID of 25 cm can occur if the image receptor is located on the contralateral side of the knee to be operated on. An OID of 5 cm can occur if the image receptor is on the ipsilateral side. However, with a 3D object, the volume it represents must also be taken into account. Therefore, one surface or point is closer to the image receptor than another surface or point. The principle of radiological identification of the MPFL femoral footprint is based on at least two references (posterior margin of the femoral diaphysis, posterior margin of the medial femoral condyle, and dorsal end of the Blumensaat line). For example, when the posterior margin of the femoral diaphysis and the femoral MPFL footprint are used as reference points, for better illustration, the distance between the two points is projected to be smaller when the image receptor is positioned medially than when it is positioned laterally, as shown in Figure 8 and Figure 9.

The position of the fluoroscopy in the operating room automatically influences the position of the X-ray tube, and thus the location of the increased scattered radiation; this plays a role in terms of the position of the surgeon for their safety [42]. Therefore, the positioning of the fluoroscopy relative to the knee is not unimportant.

This study model simulated a situation in daily clinical practice in which the femoral MPFL footprint was identified using fluoroscopic guidance. Depending on the positioning of the fluoroscopy in the operating room, the contralateral lower leg can quickly cause an increase in distance. If the image receptor is positioned close to the knee joint, as in the ML5 view, the surgeon will have more space for guidewire positioning. Conversely, if the fluoroscopy is on the contralateral side, as in the LM25 view, the surgeon will have less room to position their guidewire. Thus, they may compromise on the centralization of the knee for fluoroscopy. Meanwhile, the data of this study shows that an eccentric position does not lead to a relevant discrepancy. The size distortion was not shown to be relevant in this regard.

The results must consider several limitations. In this study, 3D-printed femora without soft tissues were used with a normal anatomic variation. An anatomic femoral MPFL footprint identification could not be established in advance, so localization of it was based on radiographic orientation, according to Schöttle et al. [23]. It is possible that the results are not generalizable to other knee variants with trochlear dysplasia, or condylar and knee malalignment, and would require further studies. However, the effect of fluoroscopy positioning in determining the same point could be demonstrated. Identification by template was determined by one author only. Thus, no intra-rater reproducibility or inter-rater reliability was determined. Other studies showed that the method applied provided high reproducibility and reliability [11,12,13]. Since printed 3D models were used, the influence of soft tissues in the identification cannot be debated. This study aimed to identify possible influences of fluoroscopy positioning when identifying the femoral MPFL point in daily practice. This study indicates that the same point has a relevant distance of approximately 5 mm when viewed from the medial position than when viewed from the lateral position, and vice versa. However, a true-lateral view projected in the outermost position shows no clinically relevant differences. This study does not give a clear preference for the position of the image receptor, which is closest to identifying the anatomical femoral MPFL footprint. Thus, there is a need for future publications to indicate the position of the C-arm in practice-relevant settings. For the moment, we recommend the LM25 view, which is the most comparable to the existing literature.

## 5. Conclusions

The present study demonstrated that the position of the image receptor of the image intensifier has a more significant influence on a precise and anatomic drill placement of the femoral tunnel in MPFL reconstruction than a centralized true-lateral radiograph. The true-lateral views in eccentric positioning for MPFL identification showed no significant difference compared to the centric position. However, a relevant difference was found between the two different positions of the image receptor visualizing the same point. This raises the question as to which side the image receptor must be positioned.

## Figures and Tables

**Figure 1 diagnostics-12-01427-f001:**
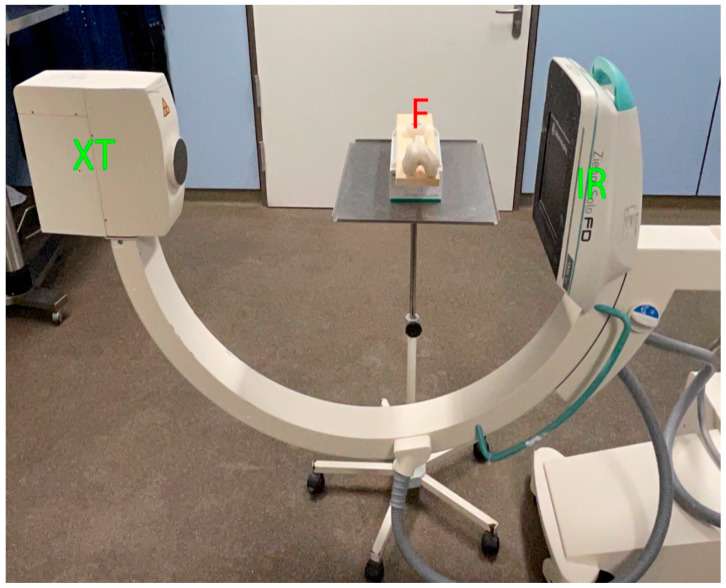
Femur positioning for the LM25 view of a right knee print. Contralateral positioning of the fluoroscopy. XT = X-ray tube, IR = image receptor, F = Femur.

**Figure 2 diagnostics-12-01427-f002:**
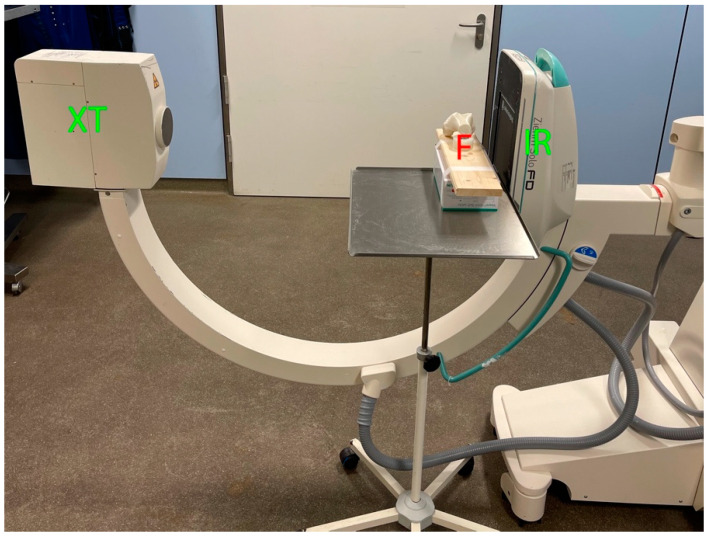
Positioning of the femur for the ML5 view of a right knee print. Ipsilateral positioning of fluoroscopy. XT = X-ray tube, IR = image receptor, F = Femur.

**Figure 3 diagnostics-12-01427-f003:**
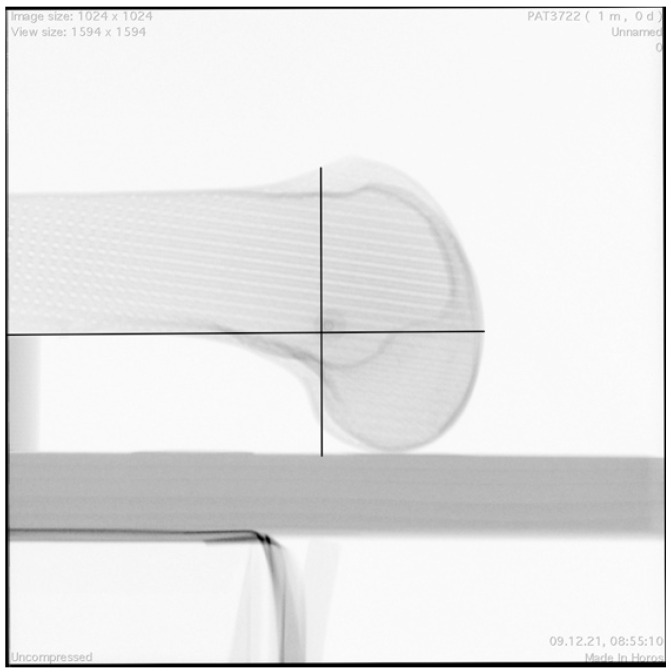
LM25 view with identified MPFL insertion point. Black lines: template according to the radiographic landmarks described by Schöttle et al. [23].

**Figure 4 diagnostics-12-01427-f004:**
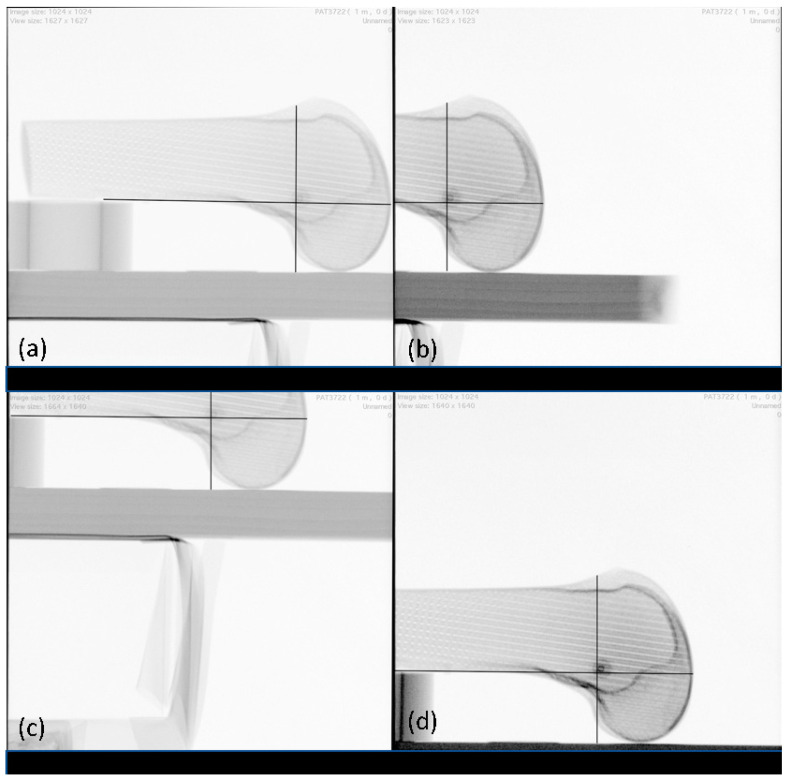
LM25 views: (**a**) distal, (**b**) proximal, (**c**) superior, (**d**) inferior. Black lines: template according to the radiographic landmarks described by Schöttle et al. [23].

**Figure 5 diagnostics-12-01427-f005:**
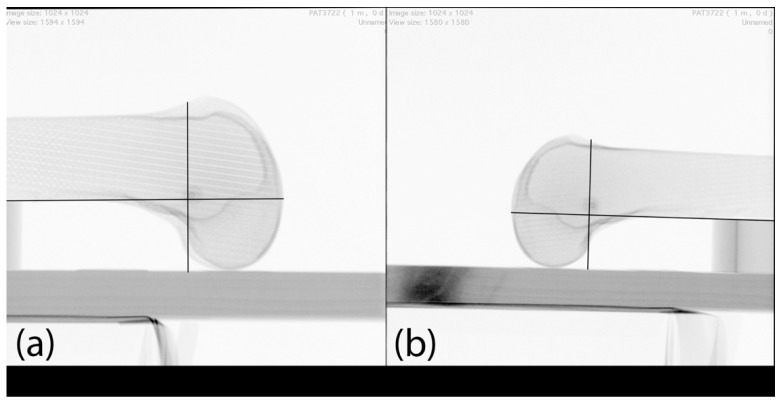
(**a**) LM25 view with identified MPFL insertion point; (**b**) the same identified MPFL insertion point controlled in ML5 view with identified MPFL insertion point. Black lines: template according to the radiographic landmarks described by Schöttle et al. [23].

**Figure 6 diagnostics-12-01427-f006:**
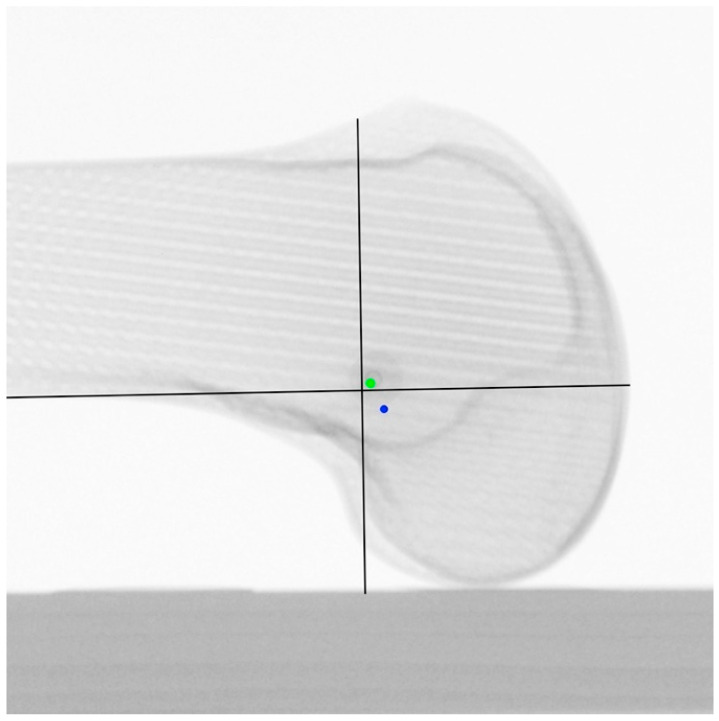
LM25 view centralized. Green point: identified MPFL insertion points from LM25 view distal, proximal, superior, and inferior. Blue point: identified MPFL insertion point from ML5 view. black lines: template according to the radiographic landmarks described by Schöttle et al. [23].

**Figure 7 diagnostics-12-01427-f007:**
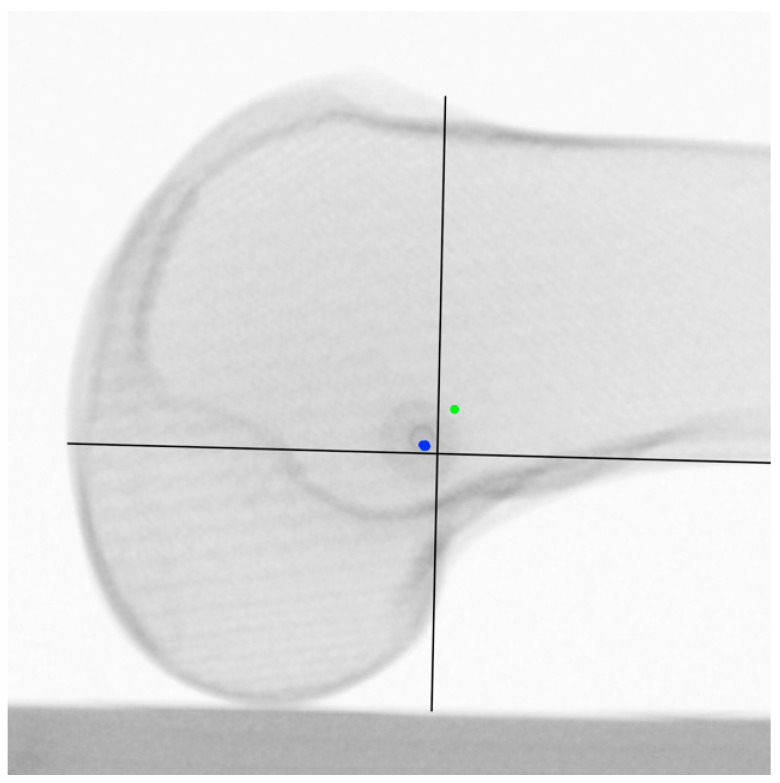
ML5 view centralized. Blue point: identified MPFL insertion points from LM25 view distal, proximal, superior, and inferior. Green point: identified MPFL insertion point from ML5 view. Black lines: template according to the radiographic landmarks described by Schöttle et al. [23].

**Figure 8 diagnostics-12-01427-f008:**
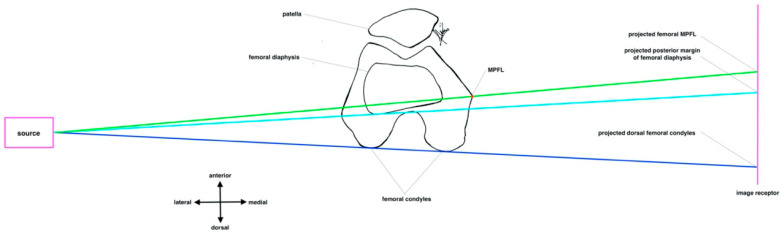
Ipsilateral positioning of fluoroscopy. Pink square: X-ray tube. Pink line: image receptor. Red point: femoral MPFL footprint. Green line: beam projecting femoral MPFL footprint. Turquoise line: beam projecting posterior margin of femoral diaphysis Blue line: beam projecting both dorsal margins of the femoral condyles.

**Figure 9 diagnostics-12-01427-f009:**
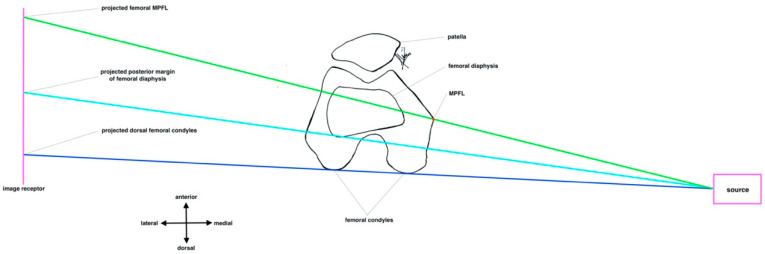
Contralateral positioning of fluoroscopy. Pink square: X-ray tube. Pink line: image receptor. Red point: femoral MPFL footprint. Green line: beam projecting femoral MPFL footprint. Turquoise line: beam projecting posterior margin of femoral diaphysis. Blue line: beam projecting both dorsal margins of the femoral condyles.

**Table 1 diagnostics-12-01427-t001:** Distances to the femoral medial patellofemoral ligament footprint.

Position	Displacement, Mean ± SD, mm	*p*-Value
LM25		
Distal	0.9 ± 0.6	0.008
Proximal	1.9 ± 1.1	0.005
Superior	1.6 ± 0.8	0.005
Inferior	1.5 ± 0.8	0.005
ML5	5.3 ± 1.2	0.005
ML5		
Distal	1.7 ± 1.3	0.008
Proximal	1.6 ± 1.2	0.005
Superior	1.3 ± 0.6	0.005
Inferior	1.6 ± 1.1	0.005
LM25	4.8 ± 2.2	0.005

## Data Availability

Not applicable.

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
