# Peer review of "Influence of the Fluoroscopy Setting towards the Patient When Identifying the MPFL Insertion Point"

_diagnostics, 2022, doi:10.3390/diagnostics12061427_

Round 1
Reviewer 1 Report
Dear Authors,
thank you for the submission. Very interesting study. I have some comments:
I believe that femoral tunnel misplacement in 31% to 67% and which leads to revision surgery in 38.2% of cases is too much. Almost the half of MPFL reconstruction are malpositioned? not possible! Please revise!
please describe the type of femur you used for the experimental set up. cadaveric? If yes, please describe the demographic of cadavers. If not, please describe the type of materials and discuss it in the discussion.
please further develop your results with those of previous studies
please further develop the limitation section
please further develop the clinical relevance and impact in the clinical practice of your findings
please increase the scientific soundness of the language level
Author Response
Dear Reviewer,
thank you very much for your comments regarding my manuscript. Here are the following corrections.
- I believe that femoral tunnel misplacement in 31% to 67% and which leads to revision surgery in 38.2% of cases is too much. Almost the half of MPFL reconstruction are malpositioned? not possible! Please revise!
Line 43-49 (105): I revised the sentence to be more clear. I referred to revision cases and not to primary MPFL-reconstruction.
“Femoral tunnel misplacement are seen in 31% to 67% of cases undergoing MPFL revision surgery [10,18,19]. A recent systematic review reported femoral tunnel misplacement as the leading reason in 38.2% of revision cases [20]. This result indicates that placement remains challenging.”
- please describe the type of femur you used for the experimental set up. cadaveric? If yes, please describe the demographic of cadavers. If not, please describe the type of materials and discuss it in the discussion.
These information were already illustrated in Line 125-135. I added the age demography and described from the beginning the type of material.
“This study used ten 3D printed polylactic acid (PLA) femurs from anonymized patients (mean 43 years old; range 17 – 69). CT scans of these patients were saved as complete DICOM series and processed using Materialise's Interactive Medical Image Control System and 3-Matic (Mimics Innovation Suite v24; 3-Matic Medical v16; Materialize, Leuven, Belgium). Post-processing was performed with global surface treatment. For the slicing process, Cura (Ultimaker Cura v4.11; Ultimaker, Utrecht, Netherlands) was utilized with a layer height of 0,1 mm and gyroid infill structure to ensure a high level of detail and steady x-ray images. 3D Printing was performed using an Ultimaker S5 Dual Head Fused Deposition Modeling (FDM) Printer with polyvinyl alcohol (PVA) as support material. None of the CT scans showed evidence of dysplasia according to the Dejour classification [22].”
- please further develop your results with those of previous studies
Lines 424 – 437:
“Several authors have already shown that not well aligned true-lateral views show slight differences in the localization of the MPFL point already at 2.5° and significant differences at 5° [16,17,34]. Howells et al. pointed out that a true lateral view with different rotations causes an error in anterior-posterior alignment [17]. Balcarek and Walde showed that a significant error could also occur with misaligned abduction and adduction [16]. They used six cadaveric femurs for this purpose and determined the femoral MPFL point according to Schöttle et al. [23]. Ziegler et al. confirmed similar results, using dissected cadaver knees with markings on each anatomical structure [34]. These studies considered neither the eccentric position of the knee joint nor the effect of OID, nor the image receptor's position. This study analyzed the relevant fluoroscopic settings for identifying the MPFL insertion point in daily practice. The eccentric positioning of the knee for identifying the femoral MPFL footprint had no clinical impact. But, the positioning of the image receptor and thus the influence of OID showed a relevant effect on the localization of the femoral MPFL footprint.”
- please further develop the limitation section
lines 626 – 645:
“The results must consider several limitations. 3D-printed femora without soft tissues were used for this study with a normal variation. An anatomic femoral MPFL footprint identification could not be established in advance, so localization of it was based on radiographic orientation, according to Schöttle et al. [23]. It is possible that the results are not generalizable to other knee variants with trochlear dysplasia, condylar and knee malalignment and would require further studies. However, the effect of fluoroscopy positioning in determining the same point could be demonstrated. Identification by template was determined by one author only. Thus, no intra-rater reproducibility or inter-rater reliability was determined. Other studies showed that the method applied provided high reproducibility and reliability [11–13]. Since printed 3D models were used, the influence of soft tissues in the identification cannot be debated. This study aimed to identify possible influences of fluoroscopy positioning while identifying the femoral MPFL point in daily practice. This study indicates that the same point has a relevant distance of approximately 5mm when viewed from the medial position than the lateral position and vice versa. However, a true-lateral view projected in the outermost position shows no clinically relevant differences. This study does not give a clear preference for the position of the Image Receptor, which is closest to identifying the anatomical femoral MPFL footprint. Thus, there is a need for future publications to indicate the position of the C-arm in practice-relevant settings. At the moment, we recommend the LM25-view, which is the most comparable to the existing literature.”
- please further develop the clinical relevance and impact in the clinical practice of your findings
The clinical relevance was further developed in different paragraphs.
First, this is the first study to our knowledge to show in a daily clinical setting that when measuring the same MPFL point, a difference of about 5mm can result from the position of the fluoroscopy. It was also shown that if the true-lateral view is not centered in the image, there is no larger scatter. Most of the previous works in the literature did not describe the distance between the knee and the receptor. However, the present work proves that this has an influence on the reproducibility of the measured values. A corresponding theoretical explanation has also been described in the text, which explains the relation of two different points of a 3D object depending on the x-ray beam. Attention is also drawn to the scattered radiation, which can be neglected during the operation.
- please increase the scientific soundness of the language level
I corrected several paragraphs to increase scientific soundness.
Best regards.
Reviewer 2 Report
This is a very interesting original article. The authors aimed to assess the sensitivity of the femoral tunnel position under lateral fluoroscopy as a function of the position of knee to the image receptor. It was a single-center study. The study is well designed, the methods and results are well presented. The discussion is interesting and well written.
Nevertheless, I have some suggestions to improve the paper:
1. The authors should decide if they want to use the abbreviation ML-5cm or ML-5. Please check through the whole text.
2. Authors should add some information about the problems with assessing the Schoettle point because of soft tissue around the knee joint in the real operation conditions.
Author Response
Dear reviewer,
thank you for your comments, I very much appreciated it.
Here are my corrections:
- The authors should decide if they want to use the abbreviation ML-5cm or ML-5. Please check through the whole text.
I changed the abbreviation to ML5 and LM25, respectively.
- Authors should add some information about the problems with assessing the Schoettle point because of soft tissue around the knee joint in the real operation conditions.
I addressed this issue in a new paragraph. Indeed, this issue is matter of discussion and reason why many surgeons prefer radiological identification while performing MPFL reconstruction.
Lines 444-453: “Rather than relying on percutaneous radiographic techniques to achieve the most anatomic MPFL reconstruction possible, an anatomic approach is suggested by some to take into account individual anatomy [34,38–40]. Thus, accurate performance of an MPFL reconstruction requires a large enough incision to identify the essential anatomic landmark, the adductor tuberosity and the medial epicondyle. This method proves more difficult in corpulent patients and could involve a large incision. Therefore, the authors recommend pre-surgical 3D CT reconstruction of the bone surface in these cases [38,41]. For daily practice, however, the Schöttle technique [23] with adjustment [16,17] has prevailed nowadays in most cases. Various studies have shown that a smaller scatter is achieved than anatomical identification [11–13].”
Round 2
Reviewer 1 Report
No additional comment. For what it may concern, the paper can be accepted. Thank you